# *Propionibacteriaium acidipropionici* CP 88 Dose Alters In Vivo and In Vitro Ruminal Fermentation Characteristics

**Jonah R. Levenson** [1], **Logan Thompson** [1], **Roderick Gonzalez-Murray** [1] , **Ryan J. Gifford** [1], **Meghan P. Thorndyke** [1], **Octavio Guimaraes** [1], **Huey Yi Loh** [1], **Briana V. Tangredi** [1], **Harrison Hallmark** [1], **Richard Goodall** [2], **John J. Wagner** [1] and **Terry E. Engle** [1,*]

[1] Department of Animal Science, Colorado State University, Fort Collins, CO 80523, USA
[2] Microbios Inc., Houston, TX 77063, USA
[*] Correspondence: terry.engle@colostate.edu

**Abstract:** Twelve 5-year-old beef steers, with an average weight of 2000 lbs., fitted with rumen canulae were used in a $4 \times 4$ incomplete Latin square design to examine the impact of the direct fed microbial Propionibacterium acidipropionici CP 88 (PA) on rumen fermentation characteristics, in vitro $CH_4$, $CO_2$, and $N_2$ production, and in vivo $CH_4$ and $CO_2$ production. All steers were housed in the same pen equipped with eight GrowSafe feeding stations to monitor individual animal feed intake and one GreenFeed System to estimate individual animal $CH_4$ and $CO_2$ production. Steers were fed a corn-silage-based diet throughout the experiment. Treatments consisted of PA administered at: (1) control (0.0); (2) $1.0 \times 10^8$; (3) $1.0 \times 10^9$; and (4) $1.0 \times 10^{10}$ cfu·animal$^{-1}$·day (d)$^{-1}$. Treatments were administered directly into the rumen as a single bolus dose daily. On day 7 and 14 of each period, rumen fluid was collected from each steer 2 h post treatment administration for VFA analysis and in vitro DM digestibility determination. Following a 14 d washout period, animal treatments were switched and the experiment repeated until the $4 \times 4$ Latin square was complete. In vivo propionic acid molar proportions and total VFA concentrations were greater ($p < 0.05$) in steers receiving PA when compared with controls. All other in vivo rumen fermentation characteristics were similar across treatments. In vitro DM disappearance ($p < 0.05$) and total VFA ($p < 0.05$) were greater and $CH_4$ lesser ($p < 0.04$) in fermentation vessels incubated with rumen fluid from animals receiving PA when compared with controls. Dry matter disappearance ($p < 0.03$) and propionic acid molar proportions increased ($p < 0.04$) linearly as the dose of PA increased. In vitro total VFA tended ($p < 0.08$) to increase linearly and $CH_4$ production per unit of DM digested tended ($p < 0.09$) to decrease quadratically in response to PA dose. All other in vitro rumen fermentation characteristics were similar across treatments. These data indicate that PA impacts in vivo and in vitro rumen fermentation characteristics.

**Keywords:** beef cattle; direct fed microbial; volatile fatty acids; methane





## 1. Introduction

Direct fed microbial (DFM) supplementation to ruminants has been reported to alter the ruminal bacterial populations and improve animal production efficiency [1]. However, the overall impacts of DFM supplementation on ruminal fermentation characteristics are not well defined. Nagaraja and others described how Propionibacterium are a lactate-utilizing bacteria, but their use as a DFM has been focused on the production of propionate, the primary glucose precursor in ruminants [2]. Fistulated steers fed a high-concentrate diet supplemented with a range of *Propionibacteria acidilactici*-DH42 doses [$1 \times 10^7$ to $1 \times 10^{10}$ cfu·animal$^{-1}$·day (d)$^{-1}$] for 7 d exhibited increases in rumen propionic acid at the expense of acetic acid across the entire range of *Propionibacteriaium acidipropionici* (strain DH42) dosages [3].

Huck and others reported that feeding beef cattle *Lactobacillus acidophilus* for 28 d and *Propionibacterium freudenreichii* for the remainder of the finishing period in feedlot cattle resulted in improved methane reduction compared with controls or *Propionibacterium freudenreichii* supplementation alone [4]. In contrast, Krehbiel [1] and others reported that feeding beef cattle *Lactobacillus acidophilus* during the entire finishing period tended to increase feedlot growth performance and carcass merit when compared with feeding *Lactobacillus acidophilus* for the first 27 d of the feeding period followed by *Propionibacterium freudenreichii* for the remainder of the 140 d finishing period [5]. Regardless of these effects on performance, limited published data exist examining the impacts of Propionibacterium species on rumen fermentation characteristics in feedlot cattle. While DFM experiments have been conducted in other species (poultry, swine, etc.) as described in a review by Krehbiel and others [1], the focus of the current experiment was to evaluate the effects of *Propionibacterium acidipropionici* (PA) on rumen fermentation characteristics and in vitro methane ($CH_4$), carbon dioxide ($CO_2$), and dinitrogen ($N_2$) and in vivo $CH_4$ and $CO_2$ production [1]. Previous research by Gifford and others indicated that PA CP 88 supplementation for beef cattle increased rumen propionate molar proportions and reduced in vitro lactic acid concentrations compared with non-supplemented controls [6]. Therefore, we hypothesized that as the dose of PA CP 88 increased, rumen fermentation characteristics would shift, in a dose-dependent manner, toward greater propionate production and less $CH_4$ production.

## 2. Materials and Methods

All animals and procedures used in this experiment were approved by the Institutional Animal Care and Use Committees at Colorado State University prior to initiating the experiment (approval number: 1875). Twelve five-year-old castrated angus beef steers, with an average weight of 2000 lbs. fitted with rumen canulae, were used in a $4 \times 4$ incomplete Latin square design to examine the impact of a DFM and PA on rumen fermentation characteristics. All steers were housed together in a dirt-surfaced pen equipped with eight GrowSafe feeding stations (GrowSafe Systems, Ltd. Calgary, AB, Canada) to monitor individual animal feed intake and one GreenFeed System (C-Lock Inc. Rapid City, SD, USA) to estimate individual animal $CH_4$ and $CO_2$ production over the course of the experiment. All steers were fed a corn-silage-based diet (Table 1) throughout the duration of the experiment. The basal diet was formulated to meet the nutrient requirements for growing feedlot cattle [7].

Following a 2-week adaptation period to the basal diet, all steers were weighed and blocked by body weight (4 steers per weight block with 3 weight blocks). Steers within a weight block were randomly assigned to one of 4 treatments. Treatments consisted of (1) control (0.0); (2) $1.0 \times 10^8$; (3) $1.0 \times 10^9$; and (4) $1.0 \times 10^{10}$ cfu·animal$^{-1}$·d$^{-1}$ of *P. acidipropionici* CP 88(PA). Immediately prior to treatment administration, rumen pH was determined by inserting a HI98190 handheld pH meter and HI12963 titanium body amplified pH electrode with a built-in temperature sensor (EcoTestr pH 2+; Oaktron 153 Instruments, Vernon Hills, IL, USA) into the rumen [6]. Following pH determination, rumen contents were thoroughly mixed by hand [6]. Appropriate dilutions of the DFM treatments were made in deionized water and administered directly into the rumen via the cannula as a single dose at 0700 h daily. For the control treatment, water and the DFM carrier were administered. The same volume of water was used to deliver all daily treatment doses.

On day 7 and 14, rumen fluid was collected from each steer 2 h post treatment administration. Approximately 250 g of rumen contents was collected from each animal and centrifuged at $28,000 \times g$ at 5 °C for 30 min. Supernatant was acidified with 25% meta-phosphoric acid and frozen at $-20$ °C until volatile fatty acid (VFA) analysis could be performed [6]. After the 14 d treatment administration, all treatments were discontinued for 14 d (washout period). Following the 14 d washout period, animal treatments were switched and the experiment repeated until the $4 \times 4$ Latin square was complete.

**Table 1.** Dry matter (DM) ingredient composition of basal diet.

| Ingredient | % |
| --- | --- |
| Corn silage | 50.0 |
| Cracked corn | 23.9 |
| Distiller's grains | 8.7 |
| Alfalfa hay | 7.2 |
| Wheat straw | 5.0 |
| Liquid supplement [1] | 4.4 |
| Limestone | 0.40 |
| Salt | 0.10 |
| Analyzed nutrient composition | |
| DM, % as fed | 62.4 |
| Crude protein, % | 15.1 |
| Acid detergent fiber, % | 17.9 |
| Neutral detergent fiber, % | 28.0 |
| Ether extract, % | 6.4 |
| NEg, Mcal/kg [2] | 1.21 |
| NEm, Mcal/kg [3] | 1.91 |
| Calcium, % | 0.63 |
| Magnesium, % | 0.22 |
| Phosphorus, % | 0.36 |
| Potassium, % | 1.41 |
| Sulfur, % | 0.24 |
| Cobalt, mg/kg | 21.8 |
| Copper, mg/kg | 18.0 |
| Manganese, mg/kg | 81.3 |
| Selenium, mg/kg | 2.1 |
| Zinc, mg/kg | 64.9 |

[1] Liquid supplement provided in a molasses suspension: 3.72% NPN (urea), 0.61% Ca ($CaCO_3$), 0.56% salt (NaCl), 2.75% K (KCl), 110,000 IU/kg vitamin A, 9.4 IU/kg vitamin E, and 440 g/metric ton of monensin (Rumensin 90, Elanco Animal Health, Greenfield, IN, USA). [2] NEg = Net energy for gain. [3] NEm = Net energy for maintenance.

During each treatment period, enteric $CH_4$ and $CO_2$ production were estimated using a GreenFeed emission measurement system (C-Lock Inc., Rapid City, SD, USA). Weekly calibrations were completed and $CO_2$ recoveries were recorded monthly [8]. Vyas and others examined the use of cannulas for estimating emissions using the sulfur hexafluoride technique and determined that gas leaking could be minimized if tight-fitting cannulas, such as the ones utilized in this experiment, were used [9]. Prior to starting the experiment, the animals were acclimated to the GreenFeed System. The GreenFeed System was then left in the pen for the entire duration of the experiment. The bait feed used in this experiment was an alfalfa pellet feed (approximately 35 g per discharge). Animals were allowed a maximum of 6 discharges per visit to the GreenFeed System with 30 s intervals between each discharge. The GreenFeed System was programmed to require 4 h between each visit and allowed each animal to visit a maximum of 6 times per day. Methane and $CO_2$ production (g/kg DMI) per animal per period were calculated by dividing total $CH_4$ and $CO_2$ production estimated by the GreenFeed System for the last 7 d in each period by the total kg of DMI consumed by each animal, as determined by the GrowSafe feeding system for the last 7 d in each period.

*2.1. Animal Husbandry*

The feeding location was checked daily to ensure that all gates were secure and that all equipment was functioning properly. Health status was monitored daily [6]. Briefly, all animals were monitored for health and locomotion problems daily. Any animal exhibiting symptoms of respiratory disease or locomotion problems was removed from the pen for a more thorough assessment by trained personnel. If an animal was determined to be moribund, the animal was treated according to the facility treatment protocol and immediately returned to their original pen. If problems persisted concerning the health

status of a specific animal, the animal was removed from the experiment. If an animal was removed from the experiment, the animal was weighed at the time of removal.

### 2.2. In Vitro Rumen Fluid Collection

As previously described, on day 7 and 14 of each period of the experiment, approximately 1 L of rumen fluid was collected from all steers 2 h post feeding [6,10]. Rumen fluid from each steer was filtered once through four layers of cheesecloth into individual pre-warmed (39 °C) thermoses. A modified McDougall's buffer solution was mixed with rumen fluid at a 1:1 ratio and pH was recorded [10,11].

### 2.3. In Vitro Chambers

Approximately 4 kg (wet weight) of the basal diet was collected from the feed truck at the time of feed mixing. The basal diet ration sample was then dried at 60 °C for 72 h in a forced air-drying oven and ground to fit through a 2.0 mm screen (Thomas Scientific, Swedesboro, NJ, USA) [6]. The ground ration was added to pre-weighed 50 mL conical tubes ($0.5 \pm 0.001$ g; 3 conical tubes per animal per incubation time point) and 100 mL glass bottles ($1.0 \pm 0.0001$ g; 3 glass bottles per animal per incubation time point). A 1:1 McDougall's buffer: rumen fluid mixture was then dispensed into all in vitro vessels (30 mL into the conical tubes and 60 mL into the glass bottles) [12]. The conical tubes were capped with one-way valves to maintain anaerobic conditions and were used to determine dry matter digestibility.

Glass bottles were capped and sealed with an airtight rubber stopper immediately after the buffer:rumen fluid mixture was dispensed into the glass bottle. The glass bottles remained sealed to maintain anaerobic conditions. The gas pressure of each glass bottle was determined, at the end of each incubation timepoint, using a digital pressure gauge (Dwyer Instruments Inc, Michigan City, IN, USA) fitted with a 20-gauge needle inserted through the rubber stopper. Gas composition ($CH_4$, $CO_2$, and $N_2$,) was determined by aspirating 10.0 mL of headspace gas from the glass bottle and immediately injecting the gas sample into the injection port of a gas chromatograph (Shimadzu GC–14A; Shimadzu, Kyoto, Japan) equipped with a thermal conductivity detector set at 100.0 °C.

All conical tubes and glass bottles were incubated in a circulating water bath at 39 °C for the appropriate lengths of time (0, 6, or 12 h) and swirled by hand every 3 h. At 0, 6, and 12 h, three tubes per animal (3 conical tubes and 3 glass bottles) were removed from the water bath. Conical tubes were centrifuged at $28,000 \times g$ at 5 °C for 30 min and glass bottles were sampled (as described above) and then uncapped and discarded. Supernatant was removed from all conical tubes and combined with 25% meta-phosphoric acid and frozen at $-20$ °C until analyzed for VFA concentrations. The remaining pellet was dried in a forced air-drying oven at 60 °C for 72 h. Following drying, the pellet dry weight was used to determine dry matter disappearance at each time point. Blank tubes containing only the McDougall's rumen fluid mixture were used to adjust the dry matter disappearance calculation for initial microbial and digesta weight contributed from the McDougall's rumen fluid mixture.

### 2.4. Volatile Fatty Acid Analysis

Post thawing, rumen fluid samples were centrifuged at $28,000 \times g$ at 5 °C for 15 min. Supernatant was analyzed for VFA composition via gas chromatography [6].

### 2.5. Statistical Analysis

A mixed effects model repeated measures analysis for a completely randomized incomplete (due to the removal of one animal from the experiment) $4 \times 4$ Latin square design was used to analyze in vivo and in vitro repeated measurements. The fixed effects were treatment, time, period, and all interactions. For all response variables measured, an individual animal or in vitro vessel was considered the experimental unit. If interactions were not significant, data were pooled and the main effects were reported. For all response vari-

ables, significance was determined at $p \leq 0.05$ and tendencies were determined at $p > 0.05$ and $\leq 0.10$. When a significant treatment or treatment $\times$ time interaction was detected, treatment means were separated using the PDIFF option of the LSMEANS statement of SAS. (SAS Inst. Inc., Cary, NC, USA). Linear, quadratic, and cubic effects were determined to examine the impact of DFM dose on rumen fermentation characteristics.

### 3. Results

One animal was removed from the experiment due to a foot injury after the second period of the experiment. All data for this animal were removed from analysis. All other animals remained healthy throughout the experiment.

The influence of live animal daily dosing of PA on in vivo rumen fermentation characteristics in fistulated steers is described in Table 2. There were no treatment $\times$ time interactions; therefore, the overall main effects are presented for all response variables. There were no treatment effects ($p = 0.89$) on daily DMI averaging 12.6 kg DM. Propionic acid concentrations and total VFA concentrations were greater ($p < 0.05$) in steers receiving DFM when compared with controls. No other fermentation characteristics differed between treatments. The influence of the in vivo daily dosing of DFM on in vitro fermentation parameters is also shown in Table 2.

**Table 2.** Influence of direct fed microbial dose on in vivo and in vitro fermentation characteristics.

| Item | Treatment [a] 0.0 [b] | 0.1 [c] | 1.0 [d] | 10.0 [e] | SEM | $p <$ Trt | Time | Trt $\times$ Time | Linear |
|---|---|---|---|---|---|---|---|---|---|
| In vivo | | | | | | | | | |
| $n$ | 11 | 11 | 11 | 11 | — | — | — | — | — |
| DMI [f], kg/d | 12.5 | 12.8 | 12.3 | 12.8 | 0.34 | 0.93 | 0.02 | 0.73 | 0.89 |
| Rumen pH, s.u. | 6.55 | 6.63 | 6.54 | 6.61 | 0.05 | 0.72 | 0.01 | 0.84 | 0.92 |
| Acetic acid, mM/100 mM | 57.1 | 57.6 | 56.2 | 56.1 | 0.97 | 0.78 | 0.08 | 0.78 | 0.46 |
| Propionic acid, mM/100 mM | 23.6 [f] | 24.8 [f] | 25.3 [g] | 26.3 [g] | 0.24 | 0.05 | 0.02 | 0.61 | 0.39 |
| Isobutyric acid, mM/100 mM | 0.51 | 0.49 | 0.47 | 0.42 | 0.09 | 0.87 | 0.01 | 0.84 | 0.18 |
| Butyric acid, mM/100 mM | 18.8 | 17.1 | 18.0 | 17.2 | 0.84 | 0.54 | 0.03 | 0.81 | 0.84 |
| Acetic acid/propionic acid | 2.4 | 2.3 | 2.1 | 2.2 | 0.21 | 0.34 | 0.06 | 0.78 | 0.19 |
| Total VFA, mM | 121.2 [f] | 125.4 [g] | 123.2 [f] | 127.1 [g] | 1.87 | 0.05 | 0.002 | 0.54 | 0.78 |
| $CO_2$, g/kg DMI | 587 | 591 | 601 | 589 | 12.3 | 0.84 | 0.01 | 0.71 | 0.21 |
| $CH_4$, g/kg DMI | 20.3 | 20.1 | 18.3 | 18.9 | 1.11 | 0.19 | 0.01 | 0.62 | 0.28 |
| In vitro | | | | | | | | | |
| DMD [g], % | 59.2 [f] | 61.7 [g] | 63.2 [h] | 65.2 [h] | 1.12 | 0.05 | 0.002 | 0.63 | 0.03 |
| Acetic acid, mM/100 mM | 50.1 | 48.2 | 49.0 | 46.7 | 1.72 | 0.88 | 0.001 | 0.91 | 0.12 |
| Propionic acid, mM/100 mM | 30.5 | 33.0 | 33.0 | 34.7 | 1.05 | 0.06 | 0.001 | 0.82 | 0.04 |
| Isobutyric acid, mM/100 mM | 1.61 | 1.65 | 1.57 | 1.67 | 0.21 | 0.87 | 0.001 | 0.78 | 0.33 |
| Butyric acid, mM/100 mM | 17.81 | 17.2 | 16.4 | 16.9 | 0.98 | 0.65 | 0.002 | 0.90 | 0.11 |
| Acetic acid/propionic acid | 1.6 | 1.5 | 1.5 | 1.3 | 0.15 | 0.23 | 0.07 | 0.58 | 0.21 |
| Total VFA [f], mM | 145.2 [f] | 147.3 [g] | 147.9 [g] | 148.7 [g] | 1.76 | 0.05 | 0.001 | 0.76 | 0.08 |
| $CO_2$, mL/g DMD | 80.2 | 81.7 | 83.1 | 81.3 | 0.41 | 0.12 | 0.0001 | 0.67 | 0.21 |
| $CH_4$, mL/g DMD | 15.56 | 13.73 | 12.69 | 12.91 | 0.11 | 0.04 | 0.0001 | 0.67 | 0.37 |
| $N_2$, mL/g DMD | 4.24 | 4.57 | 4.21 | 4.22 | 0.07 | 0.62 | 0.001 | 0.82 | 0.78 |

[a] *Propionibacterium acidipropionici* dose; [b] 0.0 cfu·animal$^{-1}$·d$^{-1}$; [c] $1.0 \times 10^8$ cfu·animal$^{-1}$·d$^{-1}$; [d] $1.0 \times 10^9$ cfu·animal$^{-1}$·d$^{-1}$; [e] $1.0 \times 10^{10}$ cfu·animal$^{-1}$·d$^{-1}$; [f] dry matter intake; [g] dry matter disappearance; [h] volatile fatty acids. [e,f,g,h] Means with different superscripts significantly differ, $p < 0.05$.

There were no treatment $\times$ time interactions for any response variables measured. Therefore, overall means are presented in Table 2. Dry matter digestibility ($p < 0.05$) and total VFA concentrations ($p < 0.05$) were greater, $CH_4$ production per unit of DM digested was lesser ($p < 0.04$), and molar proportions of propionic acid tended ($p < 0.06$) to be greater in fermentation vessels incubated with rumen fluid collected from animals receiving DFM when compared with controls. All other rumen fermentation characteristics were similar across treatments.

In vitro DM disappearance increased ($p < 0.03$) in a linear fashion as the dose of DFM increased (Table 2). The in vitro production of propionic acid increased ($p < 0.04$) and total VFA tended ($p < 0.08$) to increase linearly in response to increasing DFM dose. No

quadratic and cubic effects were detected for any of the in vivo or in vitro parameters (data are not shown).

## 4. Discussion

The results for rumen VFA obtained in the current experiment are similar to those reported previously by Gifford and others [6]. Briefly, Gifford and others utilized six fistulated steers in a crossover design to investigate the impact of PA ($1.0 \times 10^{10}$ cfu·animal$^{-1}$·day$^{-1}$) on in vivo and in vitro rumen propionic acid production and in vitro lactic acid disappearance post lactic acid dosing [6]. The authors reported that in vivo propionic acid concentrations were greater and total VFA tended to be greater in rumen fluid from steers receiving PA. In vitro total lactic acid disappearance was greater at 3 h post incubation when rumen fluid collected from animals supplemented with PA was incubated with lactic acid compared with incubation with rumen fluid collected from controls. These data indicate that *P. acidilactici* CP 88 alters rumen fermentation characteristics. Fistulated steers supplemented with *P. acidilactici*-DH42 ($1 \times 10^7$ to $1 \times 10^{10}$ cfu·animal$^{-1}$·d$^{-1}$; similar dose range to the current experiment) for 7 d/dose exhibited increases in rumen molar proportions of propionic acid and decreases in molar proportions of acetic acid across the entire range of *P. acidilactici*-DH42 dosages. When the PA was removed from the diet, butyrate production increased, which suggests that PA impacted butyrate production as well [3]. However, the authors also reported that lactic acid and rumen pH were not influenced by DFM supplementation.

In an experiment by Vyas and others [9], there were no differences in rumen pH, molar proportions of individual VFA, or total enteric $CH_4$ (g/d) production in heifers fed a 70% roughage: 30% concentrate diet when supplemented with three different Propionibacterium strains (P169, P5, P54; $5 \times 10^9$ CFU for each strain) compared with controls. Methane emission intensity expressed as g of $CH_4$ produced per kg DMI was reduced in animals receiving all three PA strains when compared with controls. This response was attributed to PA animals having a numerically greater DMI than controls. However, the authors indicated that the lack of a reduction in $CH_4$ production in PA-supplemented animals is most likely due to the inability of all three PA strains to integrate into the rumen microbiome. In the current experiment, there was no difference in DMI (38:62 concentrate-to-forage ratio) due to *P. acidipropionici* (CP 88) supplementation and no reduction in $CH_4$ emission adjusted for DMI. However, in vitro $CH_4$ production, adjusted for DMD, was decreased due to *P. acidipropionici* (CP 88) supplementation, which improved in vitro DMD. There appears to be some level of disagreement between in vivo and in vitro $CH_4$ results in the current experiment, although in vivo DMD was not determined.

Narvaez and others fed a diet of corn and corn dried distillers grain to yearling steers during the finishing period with $1.0 \times 10^{11}$ cfu·animal$^{-1}$·d$^{-1}$ of *P. acidipropionici* P169 [13]. There was no observed effect on feed intake, growth rate, feed conversion, rumen pH, total VFA production, propionate, or the acetic:propionic ratio due to *P. acidipropionici* P169 supplementation when compared with the controls. In contrast, Sanchez and others supplemented low-quality forages with a 36% CP supplement (454 g·animal$^{-1}$·d$^{-1}$) containing $6 \times 10^{10}$ cfu·animal$^{-1}$·d$^{-1}$ of *P. acidipropioni* P169 to Brangus heifers and reported that molar proportions of propionic acid were increased and the acetic:propionic ratio was reduced in animals receiving PA [14]. Lehloenya and others also supplemented *P. acidipropionici* P169 to rumen and duodenal cannulated steers fed a silage-based diet for 21 d. At a rate of $1 \times 10^{11}$ cfu·animal$^{-1}$·d$^{-1}$ of P169 [15], a trend toward increased rumen propionic acid molar proportions and a decrease in acetic acid molar proportions was observed. Collectively, these data suggest that the response to PA supplementation may be diet-dependent and/or strain-dependent. In the current experiment, VFA and $CH_4$ responses to increasing doses of PA (CP 88) were obtained in experimental diets containing approximately 40% concentrate:60% roughage with added monensin. Future research is needed to determine the impact of DFM supplementation for cattle consuming: (1) diets

containing different byproduct feeds; (2) diets containing different starch and fiber sources; and (3) diets that do not contain ionophores.

## 5. Conclusions

Under the conditions of this experiment, *Propionibacterium acidipropionici* CP 88 supplementation increased in vivo propionate acid and total VFA concentrations. In vivo dosing of *Propionibacteria acidipropionici* CP 88 also linearly increased in vitro propionic acid and dry matter disappearance. A significant effect of CP 88 treatment was also obtained for total VFA and methane production per unit of dry matter digested. In vitro reduction in methane production per unit of dry matter digested and an increase in the gluconeogenic precursor propionate indicate a possible improvement in energy efficiency for the study animals. However, the methane reduction response was only obtained in vitro. Therefore, additional in vivo research evaluating cattle feedlot performance with *Propionibacteria acidipropionici* CP 88 is warranted.

**Author Contributions:** Study conception and design were performed by J.R.L., R.G.-M. and T.E.E., J.R.L., L.T., R.G.-M, R.J.G., M.P.T., O.G., H.Y.L., B.V.T., H.H., R.G., J.J.W. contributed to data collection and manuscript preparation. Data analysis was completed by J.R.L. and T.E.E. All authors have read and agreed to the published version of the manuscript.

**Funding:** This research was supported in part by the Colorado State University Agricultural Experiment Station and by MicroBios, Houston, TX. Funding number: 5303404. The use of trade names in this publication does not imply endorsement by Colorado State University or criticism of similar products not mentioned. The mention of a proprietary product does not constitute a guarantee or warranty of the products by Colorado State University or the authors and does not imply its approval to the exclusion of other products that may also be suitable.

**Institutional Review Board Statement:** Prior to the initiation of this experiment, all animal use, handling, and sampling techniques described herein were approved by the Colorado State University Animal Care and Use Committee (Protocol #1875; appoval date: 17 May 2021).

**Informed Consent Statement:** Not applicable.

**Data Availability Statement:** Not applicable.

**Conflicts of Interest:** The funders, R. Goodall, assisted with the design of the experiment but had no role in the collection, analyses, or interpretation of data, or in the decision to publish the results.

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
