# Peer review of "Propionibacteriaium acidipropionici CP 88 Dose Alters In Vivo and In Vitro Ruminal Fermentation Characteristics"

_ruminants, doi:10.3390/ruminants2040031_

Round 1
Reviewer 1 Report
The article presented for publication in Ruminants-MDPI journal by Dr. Levenson and collaborators, mainly from the Dpt of Animal Sciences of the Colorado University, and entitled “Propionibacteria acidipropionici dose alters in vivo and in vitro ruminal fermentation characteristics”, aims the dose-dependent evaluation of a determinate strain (CP88) of P. acidipropionici on the rumen fermentation parameters and CH4 and CO2 produced in feed-lot steers
Title: I think that the specific name of the bacteria is Propionibacterium acidipropionici and not Propionibacteria. The latter is used for the plural, to refer to different species of the genus. Check this matter. Check also the italics characters use for the bacteria's scientific names and for the Latin words like “in vitro”, “in vivo”, and so on. These two aspects also apply to the body of the article text and tables.
It would probably be useful and clarifying to indicate in the title the number of strains used.
L15 It is better to say “sampled” than “removed”. Not all rumen content was obtained.
L51-53 Please check the reference to Kim et al 2000 on the actual Propionobacteria species used.
L53 and forward: use the introduced abbreviations when cited (PA, DFM… i.e.).
L71 DFM and PA abbreviations have already been defined in L 45 and L64, respectively.
L77 The basal diet was formulated “to meet or exceed”… This is too ambiguous. If the N and energy requirements were exceeded, indicate the percentage of excess. On the other hand, the diet has been declared as for growing feedlot cattle and this is difficult to assign to the 40:60 concentrate to forage formulation of this trial.
Table 1 Define the abbreviations NEg and NEm in the table footer. Net energy for growing, I suppose.
L141 It is my opinion, but I think that 1:1 buffer to rumen inoculum proportion is too high. The high concentrations of nutrients and microorganisms make it difficult to detect differences due to the experimental treatment in a maximized in vitro fermentation.
L174 It would be necessary to specify better what is the biological meaning of the study of the time factor, both for in vivo and in vitro tests.
Table 3 is almost a repetition of Table 2 with the only intention of showing the result of the orthogonal contrast analysis. In my opinion, both 2 and 3 Tables should be merged, and presented in landscape format, if necessary, to make the information more easily followed by the reader. Furthermore, in view of the lack of quadratic and cubic effects, I would avoid the presentation of such P values in the Table and it should be sufficient to explain in the text the absence of that and the observed trend to the quadratic decrease in the methane in vitro production.
When P values are lower than 0.05, post hoc differences should be indicated for the effect of the treatment, including the corresponding superscript letter row by row.
L194 It must be declared that this sentence is referred to the in vitro trial.
L229 Propionibacteria strains
L230 Indicate the concentration of the bacteria administered in the referenced literature.
L242 to L252 Check the superscript numbers (exponents) of the microbial concentration digits
L257 What was the purpose of using monensin in this experiment? In my opinion, it is a clear drawback because the full potential of the experimental intervention cannot be expressed against a Control that per se improves the studied aspects (increase in the concentration of propionic acid and inhibition of the release of enteric methane), and also for the practical use of this strategy in different countries where the use of antimicrobials as microbial growth promoters is prohibited and thus the rate of improvement in these key aspects could be more advantageous.
Conclusions:
L259 Indicate the PA strain (CP88)
L262 to L264 It is important to note that the positive effects of PA have been demonstrated under in vitro and not in vivo conditions, so this conclusion should be argued with more caution.
Reviewer 2 Report
ABSTRACT/SIMPLE SUMMARY
According to authors guideline, the front matter should contain only the abstract and no the simple summary and the abstract should be a total of about 200 words maximum.
https://www.mdpi.com/journal/ruminants/instructions
When an abbreviation/acronym/ chemical compound is written for the first time, it must be written in full
MAIN TEXT
References must be numbered in order of appearance in the text and listed at the end of manuscript.
Beware of typos and spacing errors
When an abbreviation/acronym/ chemical compound is written for the first time, it must be written in full
INTRODUCTION
Line 52 for 7 d: change d with days (d)
Line 56 performance: specify performance
Line 54-61: specify in which species the study were done
Line 64: extend the introduction with information about Propionibacteria acidipropionici with previous works also in other species, explain why authors chose Propionibacteria acidipropionici etc.
MATERIALS AND METHDS
For all equipments used add also catalogue number
Line 70: add authorization number of the experiment
Line 70: specify the breed, the sex and the age and motivate the choice. If beef were female, please give some information regarding sexual maturation and if they have ever give birth
Line 70 twelve beef: explain the statistical calculation that led to this sampling
Line 116: add a figure showing the four group and the timeline of all experiments done
RESULTS
Line 183: specify to which group belongs this animal
DISCUSSION
The discussion should be broadened by highlighting he news and the differences in the results obtained with the respect to the authors cited in the same discussion
AUTHOR CONTRIBUTION
Write following instructions for authors
https://www.mdpi.com/journal/ruminants/instructions
For research articles with several authors, a short paragraph specifying their individual contributions must be provided. The following statements should be used "Conceptualization, X.X. and Y.Y.; Methodology, X.X.; Software, X.X.; Validation, X.X., Y.Y. and Z.Z.; Formal Analysis, X.X.; Investigation, X.X.; Resources, X.X.; Data Curation, X.X.; Writing – Original Draft Preparation, X.X.; Writing – Review & Editing, X.X.; Visualization, X.X.; Supervision, X.X.; Project Administration, X.X.; Funding Acquisition, Y.Y.”,
REFERENCES
Write following instructions for authors
https://www.mdpi.com/journal/ruminants/instructions
References should be described as follows, depending on the type of work:
ï‚· Journal Articles:
1. Author 1, A.B.; Author 2, C.D. Title of the article. Abbreviated Journal Name Year, Volume, page range.
ï‚· Books and Book Chapters:
2. Author 1, A.; Author 2, B. Book Title, 3rd ed.; Publisher: Publisher Location, Country, Year; pp. 154–196.
3. Author 1, A.; Author 2, B. Title of the chapter. In Book Title, 2nd ed.; Editor 1, A., Editor 2, B., Eds.; Publisher: Publisher Location, Country, Year; Volume 3, pp. 154–196.
